



**Enhancing Extended Weather Forecasts in the TCWAGFS Model Using Deep Learning Method for SST Bias**
**Correction**
Katherine Shu-Min Li[1], Nadun Sinhabahu[1], Ben-Jei Tsuang[2], Fang-Chi Wu[1], Wan-Ling Tseng[3], Pei-Hsuan Kuo[4,5], Sying-Jyan
Wang[6], Pang-Yen Liu[4], Jen-Her Chen[4], Bin-Ming Wang[7], Yung-Yao Lan[8], Sun-Yuan Kung[9]
[1]Department of Computer Science and Engineering, National Sun Yat-Sen University, Kaohsiung, Taiwan
[2]Department of Environmental Engineering, National Chung-Hsing University, Taichung, Taiwan.
[3]Ocean Center, National Taiwan University, Taipei, Taiwan.
[4]Central Weather Administration, Taipei, Taiwan
[5]Department of Atmospheric Sciences, National Central University, Taoyuan, Taiwan
[6]Department of Computer Science and Engineering, National Chung-Hsing University, Taichung, Taiwan
[7]Critical Infrastructure Research Center, National Taipei University of Technology, Taipei, Taiwan.
[8]Research Center for Environmental Changes, Academia Sinica, Taipei, Taiwan.
[9]Department of Electrical and Computer Engineering, Princeton University, New Jersey, U.S.A.
*Correspondence to*: Ben-Jei Tsuang, btsuang@gmail.com
**Abstract.** The extended weather (> 7 days) and the seasonal climate predictions are highly dependent on the status of Madden
Julian Oscillation (MJO) and El Niño-Southern Oscillation (ENSO). Both the evolutions of MJO and ENSO are found to be
correlated to the anomalies of the global sea surface temperature (SST). To decrease the predicting SST bias (BiasSST) in a
coupled ocean-atmosphere global model, we evaluate nine well-developed machine learning algorithms. By using the Bi-
directional Long Short-Term Memory (Bi-LSTM) algorithm, it is found the bias can be reduced significantly. For example,
the Root Mean Squared Error on Day 10 forecast (denoted as D10) is reduced to 0.01 K by the Bi-LSTM algorithm while the
original bias is 0.38 K by the Taiwan Central Weather Administration Global Forecast System (TCWAGFS), of which the
error is reduced by 97%.
**1 Introduction**
Extended weather prediction, which refers to potential weather conditions up to 46 days ahead (European Centre for
Medium-Range Weather Forecasts; https://www.ecmwf.int/en/forecasts/documentation-and-support), requires an
understanding of the complex interactions between ocean, land, and atmospheric processes. The improvement of accuracy in
these predictions represents a fundamental objective within the field of meteorology, offering substantial benefits to both
society and the economy (Haiden et al., 2018). To achieve this objective, various avenues must be explored, including
increasing resolution (Demory et al., 2020; Gutjahr et al., 2016; Iles et al., 2020; Strandberg and Lind, 2021), enhancing
parameterization schemes (Lee et al., 2020; Singh et al., 2022; Su et al., 2022), and advancing air-sea coupling processes
(DeMott et al., 2014, 2015; Tseng et al., 2015, 2022; Stan, 2018; Gao et al., 2020; Lan et al., 2022, 2024). The advent of deep
learning techniques signifies a considerable leap forward in this domain, exemplifying considerable promise and, in selected
instances, rivaling the efficacy of conventional dynamic weather forecasting systems (Ji et al., 2019; Noack and Reyes, 2023;





Weyn et al., 2019). Nevertheless, despite the expansion and implementation of deep learning techniques that replace dynamic
characteristics with statistical approaches, including neural network methods (Lopatka, 2019; Schultz et al., 2021), challenges
remain, particularly in the context of extended weather forecasts. As comprehension and application of these techniques
continue to expand, they are increasingly contributing to the enhancement of extended-range predictions. Consequently, the
implementation of deep learning to enhance scheme performance or diminish model bias has emerged as a highly pragmatic
and efficacious strategy within this domain (Vitart et al., 2022). These techniques have markedly enhanced forecasts of intricate
climate phenomena such as the El Niño-Southern Oscillation (Nurdiati et al., 2021; Ham et al., 2019) and the Madden-Julian
Oscillation (Kim et al. 2021; Silini et al. 2021).

Sea surface temperature (SST) is fundamentally important for climate research and accurate simulation in numerical

weather prediction models. SST plays a crucial role in the earth's energy balance and the exchange of energy between the
ocean and atmosphere, significantly modulating sensible and latent heat fluxes across the air-sea interface. For decades,
coupled ocean-atmosphere integrations have been used for extended-range forecasts (Molteni et al., 2011; Zhou et al., 2022).
These coupled models have produced more skillful and reliable forecasts of low-frequency intra-seasonal oscillations, such as
the MJO, compared to atmosphere-only models (Kim et al., 2010). Despite the benefits of ocean-atmosphere coupling for
extended forecasts, these models also exhibit large systematic errors in SSTs, which can exceed a few degrees after four weeks.
Such errors affect atmospheric circulation and degrade forecast skill scores (Vitart & Balmaseda, 2018). Therefore, SST bias
correction has been identified as a major issue to address in coupled prediction systems (Abhilash et al., 2014, Fei et al,m 2022,
Han et al., 2022).

In recent years, data-driven approaches have gained attention in geographical and oceanic research. Compared to

physical-based numerical models, data-driven methods can automatically learn relationships with much less domain-specific
knowledge and provide prediction results from trained algorithms in a short time (Xu et al., 2023). Xiao et al. (2019) used
satellite data combined with LSTM and AdaBoost ensemble learning algorithms to predict short and mid-term daily SST. 1-
to 7- day SSTs near the Korean Peninsula were predicted to designated high water temperature events by using an LSTM
network with the European Centre for Medium-Range weather Forecasts (ECMWF) ERA5 SST time series data (Choi et al.,
2023). Similarly, SST predictions for the next five days in the East and South China Seas were made using LSTM (Jia et al.,
2022; Hao et al., 2023). However, a comprehensive evaluation of various data-driven methods for their SST bias correction
performance is currently lacking.

This study aims to address this gap by evaluating nine well-developed machine learning algorithms for their performance

in predicting SST within the TCWAGFS (Liou et al., 1997; Su et al., 2021, 2022). The proposed methodology provides a
systematic and comprehensive approach to SST prediction using time series algorithms, considering the temporal dependencies
and patterns inherent in time-series data. This approach enables the development of accurate prediction models that can
effectively forecast future SST based on past data. The output from this methodology can serve as valuable input for
climatology and oceanography studies, offering insights and immediate solutions for reducing SST bias and enhancing





prediction accuracy. The chapter is structured as follows: Section 2 describes the data, algorithms, and methodology, while
Section 3 presents the results of SST bias correction. Finally, Section 4 provides a summary of the results.
**2 Data and methodology**
**2.1 Extended weather forecast model: TCWAGFS**

The TCWAGFS, Taiwan Central Weather Administration Global Forecast System (Liou et al., 1997; Su et al., 2021,

2022), is a hydrostatic global spectral model using semi-Lagrangian advection scheme (Staniforth and Cote, 1991) with
octahedral reduced Gaussian grids ($T_{Co}$) (Hortal and Simmons, 1991; Malardel et al., 2016) in the horizontal and sigma-
pressure hybrid in the vertical. The resolution of the model is 28 km in the horizontal ($T_{Co}383$) and 72 layers with 0.1 hPa at
the model top. The model is coupled with a one-column, turbulent, and kinetic-energy-type ocean mixed-layer model (snow–
ice–thermocline, SIT) in 40°S-40°N sea grids. The 35-day SSTs from the daily runs of the TCWAGFS are analysed here. The
SIT model provides accurate SST simulation and thus facilitates realistic air–sea interaction (Tseng et al., 2022). The daily
mean SST tendencies in 40°S-60°S and 40°N-60°N sea grids are from TCWA1T (Juang et al., 2024). The bias of SST we use
for later model training are collected from every 35-day forecasts in the period of 1 January 2021 to 30 June 2022, of which
the data resolution is 2.5*2.5 degrees in whole world. The observed SST used is obtained from TCWAGFS data assimilated
systems (Su et al., 2021, 2022).
**2.2 Machine learning models**

For the machine learning model selection, nine algorithms are selected to compare their performance in predicting SST

with bias correction applied. The following machine learning methods were considered: (1) Recurrent Neural Network (RNN):
Designed to handle sequence data by incorporating memory, RNNs can retain information across time steps. However, they
often struggle with long-term dependencies due to vanishing and exploding gradient problems (Funahashi and Nakamura,
1993). (2) Autoregressive Integrated Moving Average (ARIMA): A widely used statistical method for time series forecasting,
ARIMA captures temporal dependencies through a linear combination of past values and errors (Box et al., 2015).
(3) Seasonal Autoregressive Integrated Moving Average with eXogenous regressors (SARIMAX): This model is similar to
ARIMA but includes seasonal and exogenous components, making it suitable for forecasting seasonal trends in time series
data (Vagropoulos et al., 2016). (4) Long Short-Term Memory (LSTM): A type of RNN specifically designed to handle long-
range dependencies in sequences of data, LSTMs use memory cells and gating mechanisms to learn and remember long
sequences (Hochreiter and Schmidhuber, 1997). (5) Time Series Transformer I (Prophet): An additive model developed by
Facebook for forecasting time series data, Prophet is robust to missing data and shifts in the trend, and handles outliers well
(Taylor and Letham, 2018). (6) Time Series Transformer II and III (darts Transformer and darts Exponential Smoothing):
Transformer-based models that allow parallel processing of sequence data, employing self-attention mechanisms to weigh the
importance of sequence elements (Herzen et al., 2022). (7) Time Series Imaging: Transforms time series data into image data





and applies convolutional neural networks for analysis, combining strengths of time series analysis and image processing (Li
et al., 2020). (8) Bi-directional Long Short-Term Memory (Bi-LSTM): A variant of LSTM that trains two LSTM networks on
the input sequence in different temporal orders, capturing information from both past and future states for a richer
representation of the sequence data (Schuster and Paliwal, 1997).

**2.3 Data of SST bias preprocessing**

The TCWAGFS forecast daily SST data from day 1 to day 40 is used to calculate the predicted SST bias. The predicted
SST bias, referred to as BiasSST, is defined as the difference between the TCWAGFS predicted SST on the predicted day and
the observed SST. Figure 1 shows the monthly averaged predicted SST biases in January 2021 for day 5 and day 10. The
model configuration results in two large-scale circulation gyres separated by latitude, featuring a strong meandering zonal jet.
This model emulates idealized versions of existing ocean current systems, such as the Gulf Stream in the North Atlantic and
the Kuroshio Extension in the North Pacific, both characterized by dynamic eddies interacting with a robust mean flow. As
illustrated in Figure 1, the process of predicting SST using a time series algorithm is comprehensive, beginning with rigorous
data collection and preparation. Five major areas have been selected to showcase the enhancement of bias correction, as
depicted in Fig. 1b: (a) Cold Tongue Equatorial Current, (b) El Niño Indicator Area, (c) Kuroshio Current, (d) North Atlantic
Current, and (e) Pacific Ocean Warm Pool.
(a)                                                                (b)

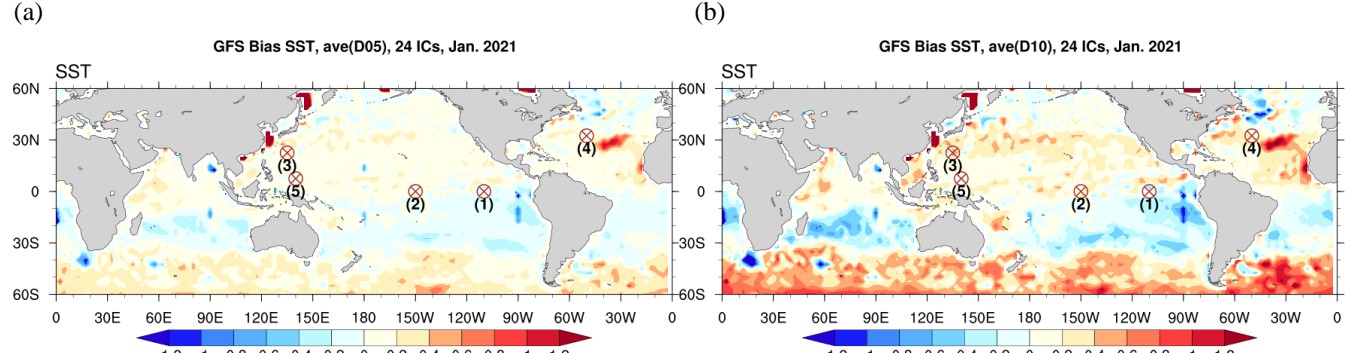

**Figure 1.** Averaged SST bias in the (a) day-5 and (b) day-10 in January 2021. The marks indicate the five selected areas. (1)
Cold Tongue Equatorial Current, (2) El Nino Indicator Area, (3) Kuroshio Current, (4) North Atlantic Current, and (5) Pacific
Ocean Warm Pool

In this study, data preprocessing is essential for maintaining the integrity of time-series patterns and ensuring that the
algorithm can learn accurate representations of sea surface temperature trends. This involves handling missing data points, a
common occurrence in real-world datasets. The technique used here is a forward-fill methodology, where null values are
replaced with the last observed non-null value. This is widely accepted, especially for time series data, as it presumes that the
absence of a data point is most closely related to the previous data point. Once the data is cleaned and complete, it undergoes



normalization using MinMaxScaler, a popular feature scaling technique. This scales all data values to a range between 0 and
1, ensuring no individual feature disproportionately influences the model due to its scale. Additionally, feature engineering is
often applied to enhance model performance. This involves creating new features from existing data, such as polynomial
features, to help uncover complex patterns within the data. Dimensionality reduction techniques like Principal Component
Analysis (PCA) are also used to reduce the number of features in the dataset while retaining most of the important information,
making the model more efficient and less prone to overfitting. The way data is represented or framed for a machine-learning
algorithm can significantly impact model performance. With the data split, the next step is to preprocess it to fit the needs of
the time series algorithm. Using a technique called the sliding window approach, sequences are generated from the data for
the time series algorithm. The size of the window is defined by the 'look_back' parameter, which determines how many
previous time steps the algorithm should consider when making a prediction. Additionally, the 'time_steps' parameter decides
how many steps into the future the algorithm should predict.
**2.4 Algorithm development**
Following data collection and preprocessing, nine time-series algorithms (RNN, ARIMA, SARIMAX, LSTM, Prophet,
darts Transformer, darts Exponential Smoothing, Time Series Imaging, Bi-LSTM) were trained using the Adam optimizer,
with an initial learning rate of 0.001. Early stopping was applied based on validation loss, with a patience threshold of 250
epochs. Dropout layers and L2 regularization were added to prevent overfitting. A 5-fold cross-validation strategy was used
to ensure robustness of the results. The predicted BiasSST was then applied to adjust the TCWAGFS SST forecasts. According
to the results, the bidirectional LSTM algorithm is selected based on the algorithm performance to make predictions. LSTM is
a type of recurrent neural network (RNN) that is particularly well-suited to time series prediction tasks like this one because
of its ability to learn long-term dependencies and its capacity to consider both past and future data in the algorithm. The
algorithm structure consists of an input layer, followed by a Bidirectional LSTM layer, and finally, a fully connected layer that
maps the LSTM outputs to the final output size. The LSTM layer has a specific number of hidden units and LSTM layers,
which are parameters that can be tuned to enhance the algorithm's performance. Training the algorithm involves optimizing
its parameters to minimize a given loss function. The algorithm employs the Adam optimizer, an algorithm for first-order
gradient-based optimization of stochastic objective functions. This is paired with the Mean Squared Error (MSE) loss function
which calculates the quantity that the algorithm seeks to minimize during training.
One of the strengths of this algorithm is its capacity to look both backward and forward in time due to the use of a
bidirectional LSTM. It uses past data (going back a specific number of steps defined by the 'look_back' parameter, which is
set as 1 in our work) and future data (defined by the 'time_steps' parameter, which is set as 3 in our work) to make a prediction.
A sensitivity analysis was conducted by varying the look-back window size from 1 to 5 days, and the Bi-LSTM performance
was evaluated for each configuration. The model showed optimal performance with a look-back window of 3 days, beyond
which performance gains were negligible. This analysis demonstrated the robustness of the Bi-LSTM model for time-series
SST predictions.





For each data source (BiasSST), we calculated the following error metrics: Root Mean Square Error (RMSE), Mean
Absolute Error (MAE), and Mean Bias Error (MBE). These three-error metrics can be calculated by following Eqs. (1)-(3):
$RMSE = \sqrt{\frac{1}{n}\sum_{i=1}^{n}(O_i - P_i)^2},$                                            (1)
$MAE = \frac{1}{n}\sum_{i=1}^{n}|O_i - P_i|,$                                              (2)
$MBE = \frac{1}{n}\sum_{i=1}^{n}(O_i - P_i),$                                              (3)
where $n$ is sample size, $P_i$ is the prediction and $O_i$ is the true value.
The algorithms were implemented using Python 3.11.2 and PyTorch 2.0.1. To enhance performance, noise removal and
optimization techniques were applied, including data smoothing, the Huber loss function, overfitting compensation, and post-
training fairness calculations. The training process consists of three main stages: pre-training, algorithm training, and post-
training. During the pre-training stage, data preprocessing begins with applying a moving average to reduce noise. The data is
then split into training, validation, and test sets, using a 75%/15%/10% ratio, respectively. In the training stage, MSE is used
as the loss function, and the Adam optimizer is applied with a learning rate of 0.001. After training, the main algorithm
generates an initial prediction, which is further refined by the secondary algorithm to produce the final, optimized prediction.

**3 Experimental Results**
**3.1 Performance of Bi-LSTM for BiasSST Prediction**
Table 1 reports the performance of Bi-LSTM on day 10 with and without noise removal and optimization. Algorithms
that incorporate noise removal and overfitting prevention generally show improved performance.



**Table 1.** Performance of Bi-LSTM for BiasSST prediction on day 10 (without noise removal and overfitting prevention) without and with noise removal. The 5 selected regions are marked in Fig. 1b.

| Region | | Metrics (unit:K) | w/o noise removal | w/ noise removal |
|---|---|---|---|---|
| **(1)** | **Cold Tongue Equatorial Current** | **RMSE** | 0.044 | 0.008 |
| | | **MAE** | 0.040 | 0.006 |
| | | **MBE** | 0.040 | 0.006 |
| **(2)** | **El Niño Indicator Area** | **RMSE** | 0.009 | 0.002 |
| | | **MAE** | 0.007 | 0.002 |
| | | **MBE** | -0.007 | -0.001 |
| **(3)** | **Kuroshio Current** | **RMSE** | 0.035 | 0.013 |
| | | **MAE** | 0.031 | 0.006 |
| | | **MBE** | 0.027 | -0.004 |
| **(4)** | **North Atlantic Current** | **RMSE** | 0.014 | 0.006 |
| | | **MAE** | 0.013 | 0.005 |
| | | **MBE** | 0.001 | 0.004 |
| **(5)** | **Pacific Ocean Warm Pool** | **RMSE** | 0.019 | 0.004 |
| | | **MAE** | 0.017 | 0.003 |
| | | **MBE** | -0.017 | -0.002 |
| | **Average** | **RMSE** | 0.024 | 0.007 |
| | | **MAE** | 0.022 | 0.004 |
| | | **MBE** | 0.009 | 0.001 |

### 3.2 Identify the improvement of Bi-LSTM

The TCWAGFS is the foundational model used for predicting sea data statistics. It is developed based on established physical laws and relationships governing oceanic and atmospheric conditions. This model relies on equations derived from physics to forecast future states of the sea, including factors such as temperature, currents, and sea level pressure. However, it is limited by its simplified assumptions and inability to fully capture the complexities and variabilities of natural processes. This limitation becomes evident in the model's progressively increasing RMSE, MAE, and MBE values over the course of 30 days. While the Bi-LSTM is suggested as the best algorithm to correct the SST bias, the further evaluation is presented here. The performance metrics provide key insights into the magnitude, direction, and percentage of error relative to the actual data, helping to evaluate the model's accuracy. As shown in Figure 2, different algorithms demonstrate varying levels of effectiveness in predicting SST in the El Niño indicator area and the Western Pacific warm pool. These predictions are evaluated based on the bias temperature gradient (BiasSST) data collected over 30 days. This information is critical for understanding and forecasting climate phenomena such as El Niño.

Focusing first on the TCWAGFS model, the RMSE values for the El Niño indicator area increase from 0.129 to 0.591 over 30 days, while in the Western Pacific warm pool, the RMSE values rise from 0.169 to 0.453. This upward trend indicates a growing magnitude of prediction errors over time, suggesting that the model's performance deteriorates as the forecast





horizon extends. A similar pattern is observed with the MAE and MBE metrics, further supporting this conclusion. In contrast,
the Bi-LSTM algorithm, a recurrent neural network variant, consistently achieves lower RMSE, MAE, and MBE values
compared to the TCWAGFS model. Bi-LSTM is particularly well-suited for tasks involving sequential data because it
processes information in both directions—backwards and forwards—allowing it to understand the context of data more
effectively. This ability makes Bi-LSTM especially effective in time-series predictions where surrounding context is crucial
for improving accuracy. Similar performance improvements are observed in other locations shown in Figure 1b (not depicted
here). Several factors explain why the Bi-LSTM algorithm outperforms traditional models in sea data prediction, a time-series
problem: (1) Sea data often has long-term dependencies. Traditional methods may struggle to capture these over extended
periods. For example, sea currents or temperatures may be influenced by factors from days, weeks, or even months earlier.
LSTM's ability to learn and retain such long-term dependencies is advantageous, and Bi-LSTM enhances this by analyzing
data in both directions. (2) Unlike standard LSTMs, which process data from past to future, Bi-LSTMs look at data in both
directions. This is particularly beneficial in cases where future data provides additional context to past states, leading to more
accurate predictions. For instance, knowing about a future storm can inform the prediction of current sea conditions. (3) Noise
Reduction: Sea data often contains noise due to environmental variability, sensor errors, or other factors. Bi-LSTM's
bidirectional processing helps it manage this noise more effectively, improving prediction accuracy. (4) Complex Patterns:
Sea data can exhibit intricate temporal patterns due to factors like tidal movements, seasonal shifts, and long-term climate
trends. Bi-LSTM's architecture, which captures both linear and non-linear relationships, is well-suited for learning and
predicting such patterns. (5) Robustness: Sea data can be volatile, with sudden changes due to extreme weather events, such
as storms or tsunamis. Bi-LSTMs are robust enough to handle these sudden shifts, maintaining strong prediction performance
even under such conditions.

The key reason Bi-LSTMs perform better in this context is their ability to incorporate both past and future information

into predictions, giving the model a more holistic view of the data. This leads to more accurate and informed predictions,
particularly when dealing with complex, temporal, and noisy datasets like sea data statistics. Bi-LSTM's performance suggests
that machine learning methods, particularly those capable of handling long-range dependencies and complex patterns, have
significant potential to outperform traditional physical models in capturing the complexities of environmental systems.





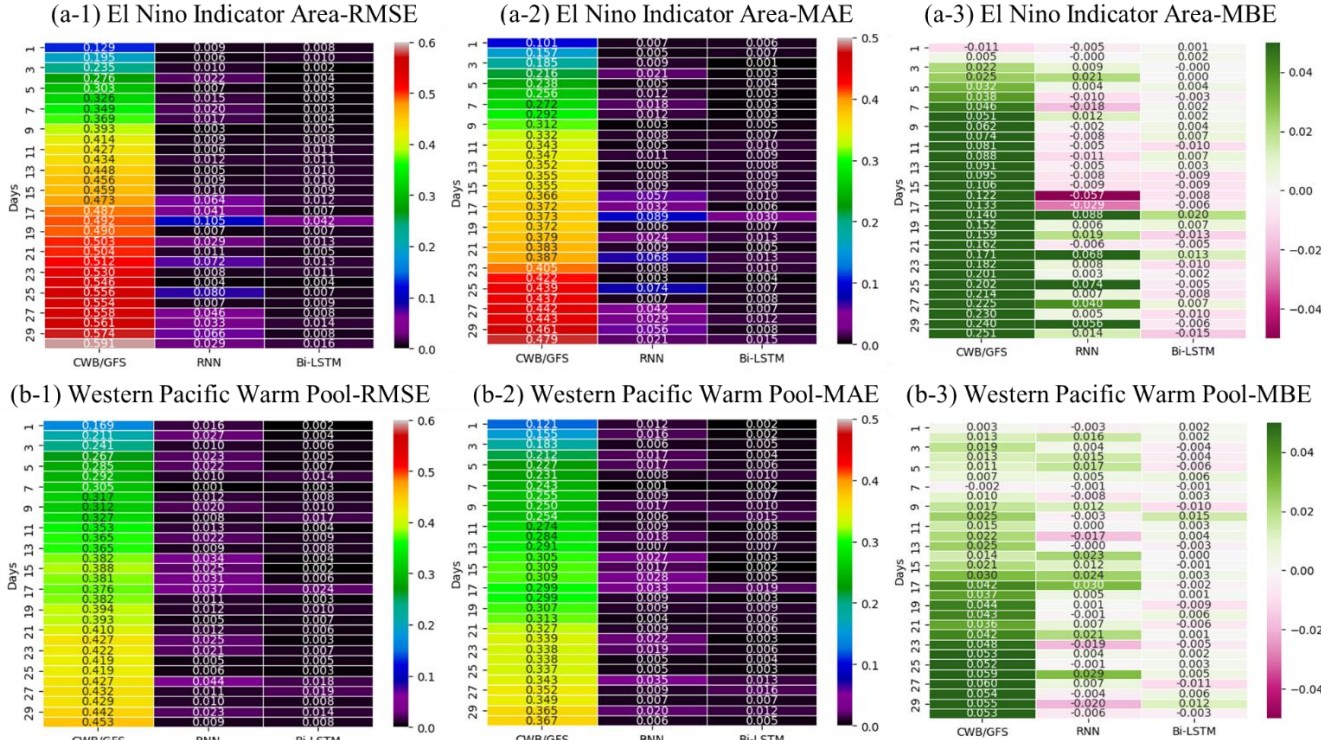

**Figure 2.** Variation of performance parameters over 30 days for BiasSST in two selected areas: (a) El Niño indicator area (Mark 2 in Figure 1b) and (b) Western Pacific warm pool (Mark 5 in Figure 1b).

The simple RNN method generally produces RMSE, MAE, and MBE values that are higher than those of the Bi-LSTM algorithm but still lower than those of the TCWAGFS model. This suggests that while the RNN can process time-series data effectively, it is not as sophisticated or efficient as the Bi-LSTM. The performance of the three algorithms from day 1 to day 5, day 10, day 15, day 20, day 25, and day 30 is summarized in Figure 2. Both RNN and Bi-LSTM consistently outperform the TCWAGFS model. For day 5, the Bi-LSTM algorithm achieves the best results with the lowest RMSE (0.005 K) and MAE (0.004 K), while RNN also outperforms the TCWAGFS model, though not as well as Bi-LSTM. As the forecast horizon extends to day 10, Bi-LSTM remains the best performer, with an RMSE of 0.008 K and an MAE of 0.007 K. The RNN algorithm shows a slight increase in errors compared to day 5 but continues to perform better than the TCWAGFS model. By day 15, Bi-LSTM still leads with an RMSE of 0.009 K and an MAE of 0.009 K, although it slightly underestimates forecasts, as indicated by its negative MBE (-0.009 K). On day 20, Bi-LSTM experiences a moderate increase in RMSE (0.013 K), though it remains the top performer. By day 25, Bi-LSTM improves again with an RMSE of 0.007 K and an MAE of 0.007 K, while RNN shows increased errors. Finally, by day 30, Bi-LSTM still leads with an RMSE of 0.016 K and an MAE of 0.015 K, outperforming the other algorithms. RNN sees a slight reduction in RMSE and MAE compared to day 25 but is still inferior to Bi-LSTM.





The TCWAGFS model struggles to predict complex and volatile sea data statistics over extended periods. Meanwhile,
both the RNN and Bi-LSTM algorithms provide promising alternatives due to their ability to process sequential data. RNN
offers improvements over traditional mathematical models, as recurrent neural networks are designed for sequential data and
can recognize patterns over time. However, RNN's performance is limited by issues like vanishing and exploding gradients,
which make it difficult for the model to capture long-term dependencies. In contrast, Bi-LSTM outperforms both the
TCWAGFS and RNN algorithms. Its ability to process data in two directions—past to future and future to past—provides
additional context, improving prediction performance. Bi-LSTM also proves to be more robust against the inherent noise and
volatility in sea data, making it a more reliable choice for such predictions. However, while Bi-LSTM demonstrates strong
results, it is not without limitations. Its performance can be highly dependent on the selection of hyperparameters, and finding
the optimal set can be computationally intensive. Additionally, Bi-LSTM, like other deep learning algorithms, requires a large
amount of training data, which may not always be available. The BiasSST prediction performance of GFS, SARIMAX, and
Bi-LSTM is shown in Figure 3. As illustrated, the Bi-LSTM method outperforms the others, as it successfully predicts the
systematic error (Column: System Error), leaving only random noise (Column: Random Noise). The selected-days forecasting
performance for BiasSST of GFS and Bi-LSTM is also summarized in Table 2, where Bi-LSTM consistently achieves the best
performance, with lower RMSE, MAE, and MBE values. The average RMSE for BiasSST over days 5, 10, 15, 20, and 30 in
the sea grids (0E-360E, 60S-60N) for TCWAGFS and Bi-LSTM is 0.666 K and 0.009 K, respectively. Under the same
conditions, the MAE is 0.559 K for TCWAGFS and 0.033 K for Bi-LSTM, while the MBE is 0.096 K and 0.0005 K,
respectively. Figure 4 highlights the bias correction effect from Day 5 to Day 30. The left panels display the BiasSST of the
TCWAGFS model, and the right panels show the bias after correction, which is the BiasSST of the TCWAGFS model minus
the bias predicted by Bi-LSTM. For example, on Day 5, the average bias for the TCWAGFS model is 0.0261 K, and after bias
correction, it is reduced to -0.009 K. In this case, the bias is reduced to just 3.4% of the original value, the highest reduction
observed. On average, the bias shown in Figure 4 is reduced to around 2%. This means that, after bias correction, only minimal
residual bias remains, resulting in more accurate SST predictions. The improvements in SST bias reduction, as measured by
MAE (Figure A1) and RMSE (Figure A2), are provided in the Appendix, further demonstrating the effectiveness of the
BiasSST correction.


(a-1)                                    (a-2)                                    (a-3)

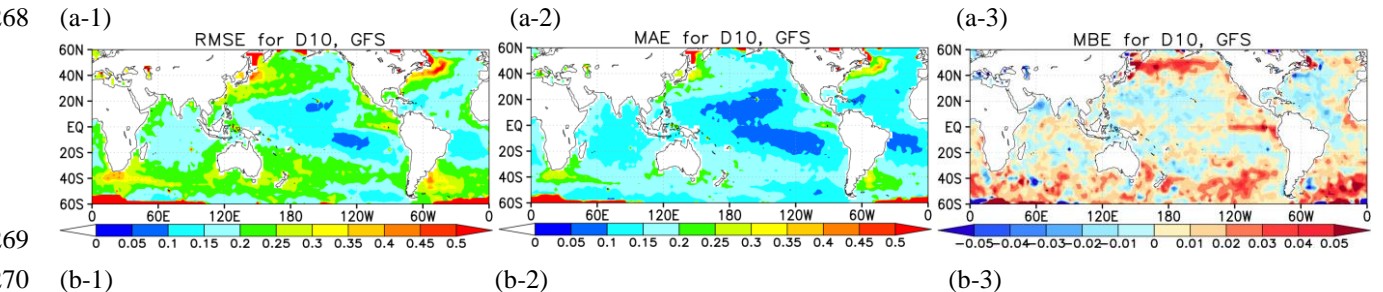


(b-1)                                    (b-2)                                    (b-3)



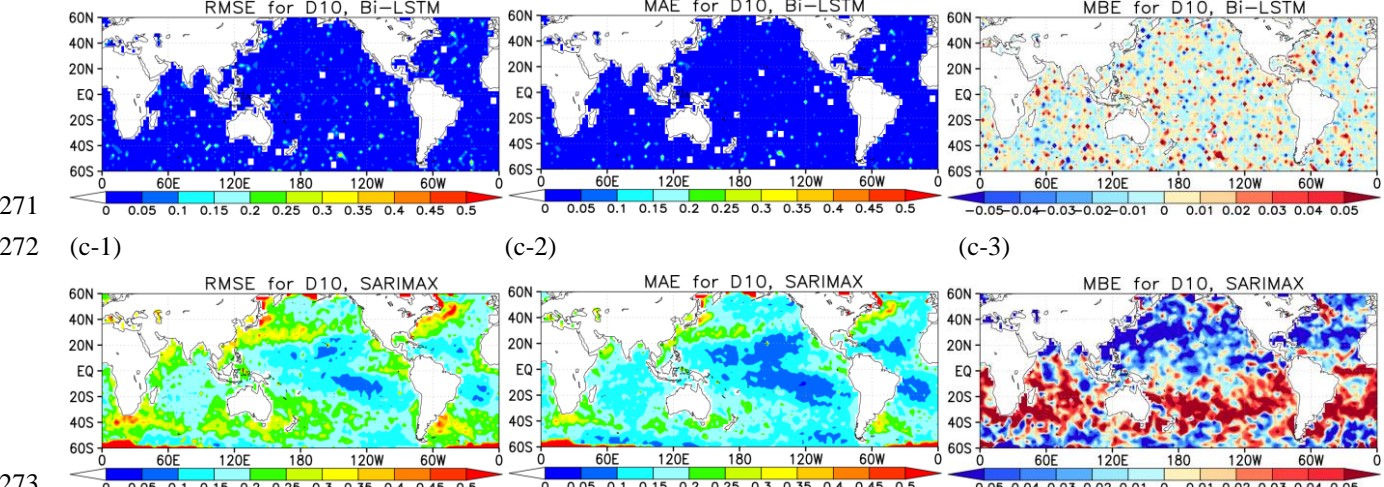

**Figure 3** BiasSST prediction performance for validation data (15%) of (a) GFS, (b) Bi-LSTM, and (c) SARIMAX on day 10
for RMSE (the left column), MAE (the middle column), and MBE (the right column).

**Table 2.** Selected-days forecasting performance for BiasSST of selected algorithms in sea grids of (0°E-360°E, 60°S-60°N).

| Day | Model | RMSE (K) | MAE (K) | MBE (K) |
|---|---|---|---|---|
| 5 | TCWAGFS | 0.223 | 0.307 | 0.031 |
| | Bi-LSTM | 0.009 | 0.015 | -0.001 |
| 10 | TCWAGFS | 0.379 | 0.414 | 0.057 |
| | Bi-LSTM | 0.010 | 0.023 | -0.001 |
| 15 | TCWAGFS | 0.547 | 0.508 | 0.082 |
| | Bi-LSTM | 0.007 | 0.030 | -0.001 |
| 20 | TCWAGFS | 0.739 | 0.607 | 0.108 |
| | Bi-LSTM | 0.009 | 0.037 | 0.002 |
| 25 | TCWAGFS | 0.928 | 0.706 | 0.137 |
| | Bi-LSTM | 0.010 | 0.043 | 0.003 |
| 30 | TCWAGFS | 1.182 | 0.814 | 0.158 |
| | Bi-LSTM | 0.011 | 0.048 | 0.001 |
| Average | TCWAGFS | 0.666 | 0.559 | 0.096 |
| | Bi-LSTM | 0.009 | 0.033 | 0.0005 |






(a-1) Average bias = 0.031 K       (a-2) Average bias = -0.001 K

(b-1) Average bias = 0.057 K       (b-2) Average bias = -0.001 K

(c-1) Average bias = 0.108 K       (c-2) Average bias = 0.002 K

(d-1) Average bias = 0.158 K       (d-2) Average bias = 0.001 K

**Figure 4.** The bias correction effect of the model results. The left column represents the original bias from GFS, while the
right column displays the bias correction after applying the Bi-LSTM method. The figures are arranged in chronological order
from top to bottom, covering Day 5 (a-1, a-2), Day 10 (b-1, b-2), Day 20 (c-1, c-2), and Day 30 (d-1, d-2).





## 4 Conclusion


This study evaluates the performance of nine well-established machine learning algorithms for predicting sea surface
temperature (SST) in the TCWAGFS model, offering a systematic and comprehensive approach to time-series-based SST
forecasting. The algorithms considered include Recurrent Neural Network (RNN), Autoregressive Integrated Moving Average
(ARIMA), Seasonal ARIMA with exogenous variables (SARIMAX), Long Short-Term Memory (LSTM), Prophet, Darts
Transformer, Darts Exponential Smoothing, Time Series Imaging, and Bidirectional Long Short-Term Memory (Bi-LSTM).
Compared to traditional dynamical models like TCWAGFS, these machine learning approaches excel due to their ability to
handle sequential data, capture complex temporal dependencies, and learn patterns from past time steps, which is crucial for
accurate weather and climate predictions. Among the algorithms tested, Bi-LSTM stands out as the most effective, consistently
outperforming the others, including conventional methods like RNN and the baseline TCWAGFS model. Bi-LSTM achieved
significantly lower RMSE, MAE, and MBE values, demonstrating its superior ability to predict SST bias. The bidirectional
structure of the Bi-LSTM model, which processes information in both forward and backward directions, allows it to capture
long-term dependencies and contextual information critical to the accurate prediction of time-series data, such as SST statistics.
This capability positions Bi-LSTM as a powerful tool for improving the accuracy of SST bias correction in the TCWAGFS
model.
A critical takeaway from this study is the importance of accounting for temporal dependencies in SST prediction.
SSTs are influenced by complex, historical factors that have long-lasting effects on present conditions. Conventional methods
often struggle to retain and leverage these long-term dependencies, which limits their predictive accuracy. Bi-LSTM, by
contrast, excels in learning and retaining these intricate relationships, providing a more holistic and accurate view of SST
evolution. The comparison between the original bias from TCWAGFS and the bias correction achieved through the Bi-LSTM
method illustrates the algorithm's ability to reduce the original bias by an average of 98%. This substantial improvement
highlights the potential for machine learning methods to enhance the precision and reliability of environmental forecasts. While
the offline bias correction achieved by Bi-LSTM in this study is significant, there is even greater potential in integrating
dynamical and statistical models for real-time applications. A hybrid approach that combines the strength of dynamical models,
like TCWAGFS, with the advanced learning capabilities of machine learning models could further enhance extended weather
forecasts. Such an integration could address not only SST bias but also other biases within global forecasting systems, offering
a more comprehensive solution for long-range and sub-seasonal weather prediction. Looking ahead, the next step will involve
transitioning from offline bias correction to the full integration of the Bi-LSTM model with the TCWAGFS system. This
integration will aim to reduce SST bias in real-time forecasting, thereby improving the overall accuracy of the TCWAGFS
model in predicting weather and climate conditions. Additionally, this approach could be extended to other climatological
variables that exhibit similar temporal dependencies, such as atmospheric temperature, humidity, and precipitation patterns.
The broader implications of this study extend beyond SST bias correction. The machine learning algorithms explored here,
particularly Bi-LSTM, demonstrate a significant potential for addressing challenges in various domains of environmental and



climate modeling. Their capacity to handle sequential data and capture long-term dependencies offers valuable insights not only for improving numerical weather prediction models but also for enhancing the accuracy of sub-seasonal and seasonal climate predictions. Moreover, the methodologies developed in this study could be adapted to other domains of Earth system science, where bias correction and long-range forecasting are essential for policy-making, disaster management, and sustainability planning.

In summary, this study underscores the transformative potential of machine learning algorithms in environmental forecasting. The superior performance of Bi-LSTM in reducing SST bias, coupled with its broader applicability in time-series analysis, makes it a valuable tool for improving the accuracy and reliability of numerical weather prediction models. By incorporating advanced statistical techniques alongside traditional dynamical models, we can enhance the precision of future climate and weather forecasts, ultimately contributing to more informed decision-making in the face of environmental challenges.

**Code and data availability**

All model codes and data presented in this study are archived at Zenodo: https://doi.org/10.5281/zenodo.15400607 (Li et al., 2025). The development repository is also publicly accessible via GitHub: https://github.com/btsuang/CWA-biastg-Train. All experiments were conducted at the National Center for High-Performance Computing.

**Acknowledgements**

This work was supported by the Taiwan Ministry of Science and Technology (grant nos. MOST 111-2111-M-002-015-, MOST 112-2121-M-005-001- and MOST 113-2121-M-005-006-). We are grateful to the National Center for High-Performance Computing for providing computer facilities.

**Author contributions**

SML, NS, FCW, WLT, PHS and SJW conceptualized the study, performed data analysis, and wrote the manuscript. BJT, PYL, JHC, YYL, and SYK developed the model and conducted the simulations. All authors reviewed and approved the final manuscript.

**Competing interests**

The contact author has declared that none of the authors has any competing interests.

**Financial support**

This research has been supported by the Ministry of Science and Technology, Taiwan (grant nos. MOST 111-2111-M-002-015-, MOST 112-2121-M-005-001- and MOST 113-2121-M-005-006-).



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

**Appendix**

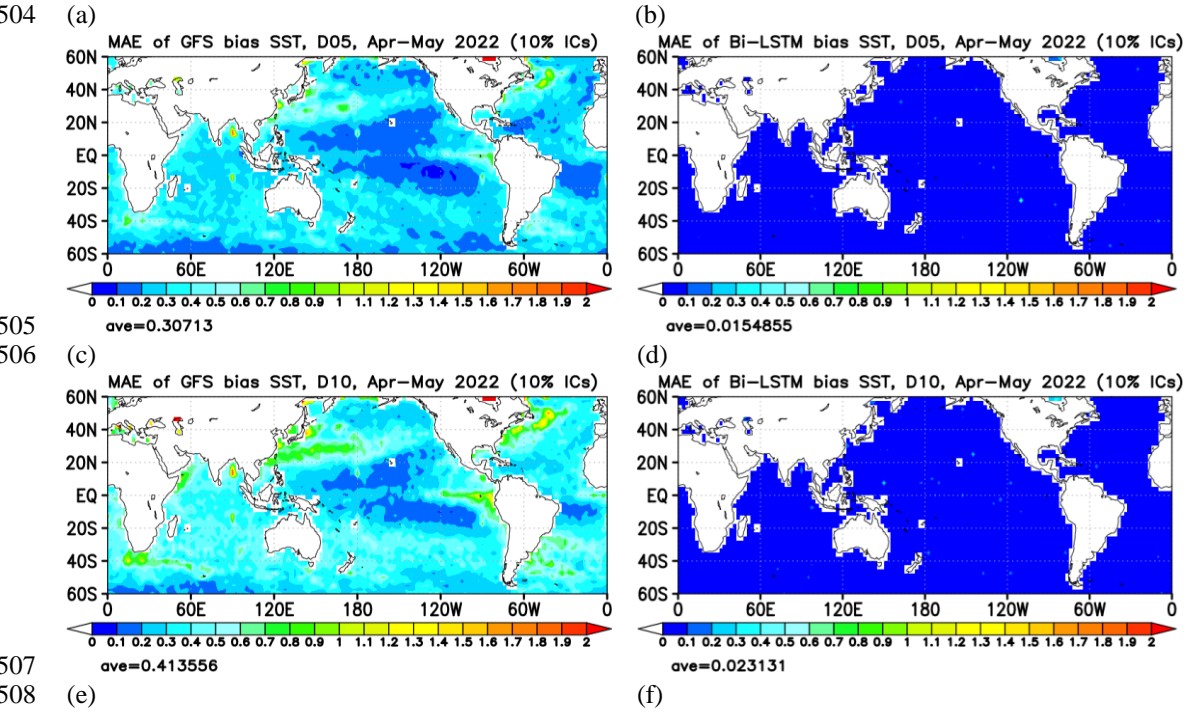



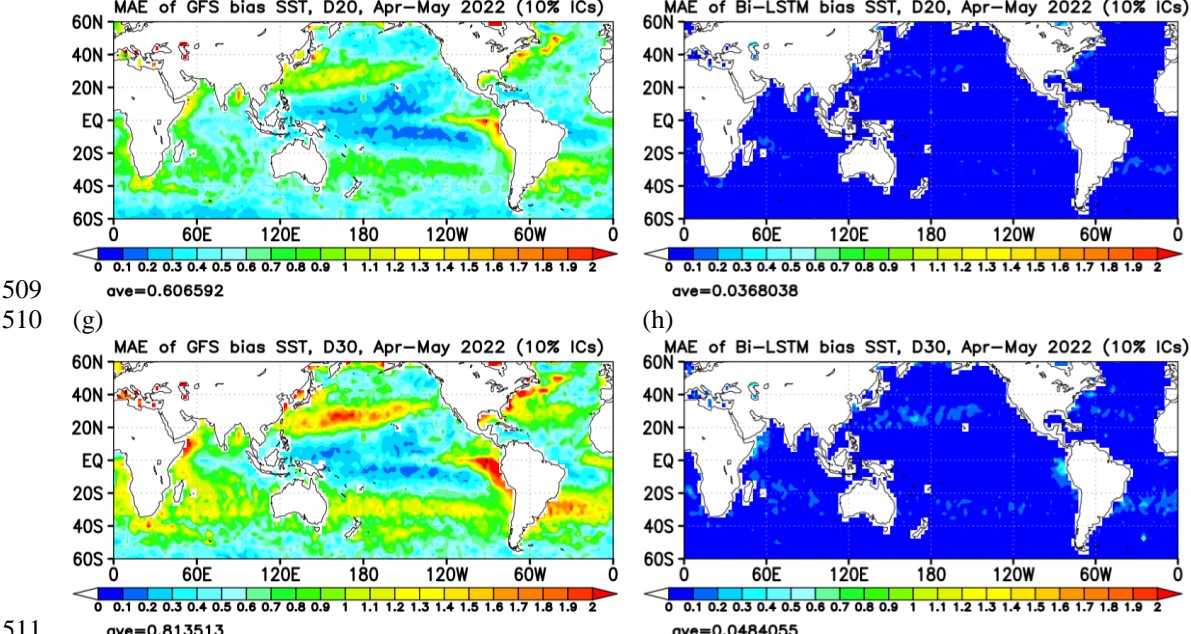

(g)                                                    (h)

**Figure A1.** The forecasting performance (MAE) for BiasSST of the model results. The left column represents the MAE results
from GFS, while the right column displays the MAE results from Bi-LSTM. The figures are arranged in chronological order
from top to bottom, covering Day 5 (a, b), Day 10 (c, d), Day 20 (e, f), and Day 30 (g, h).


(a)                                                    (b)

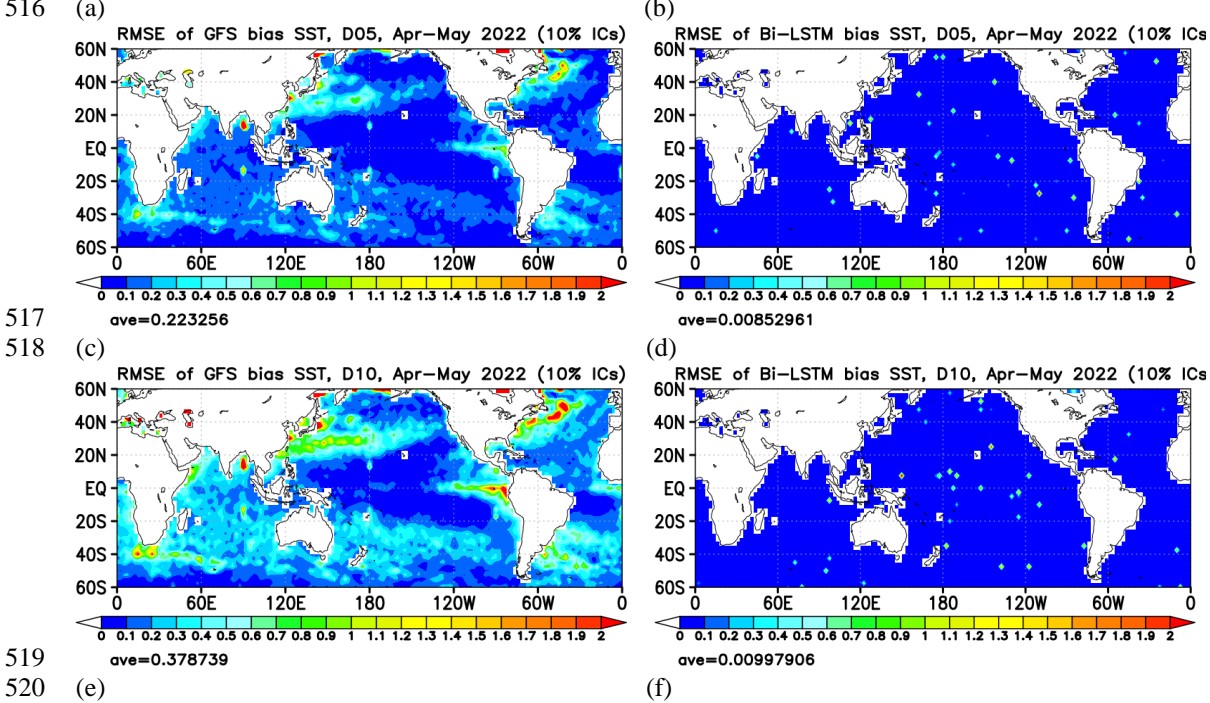

(c)                                                    (d)
ave=0.378739                                           ave=0.00997906
(e)                                                    (f)





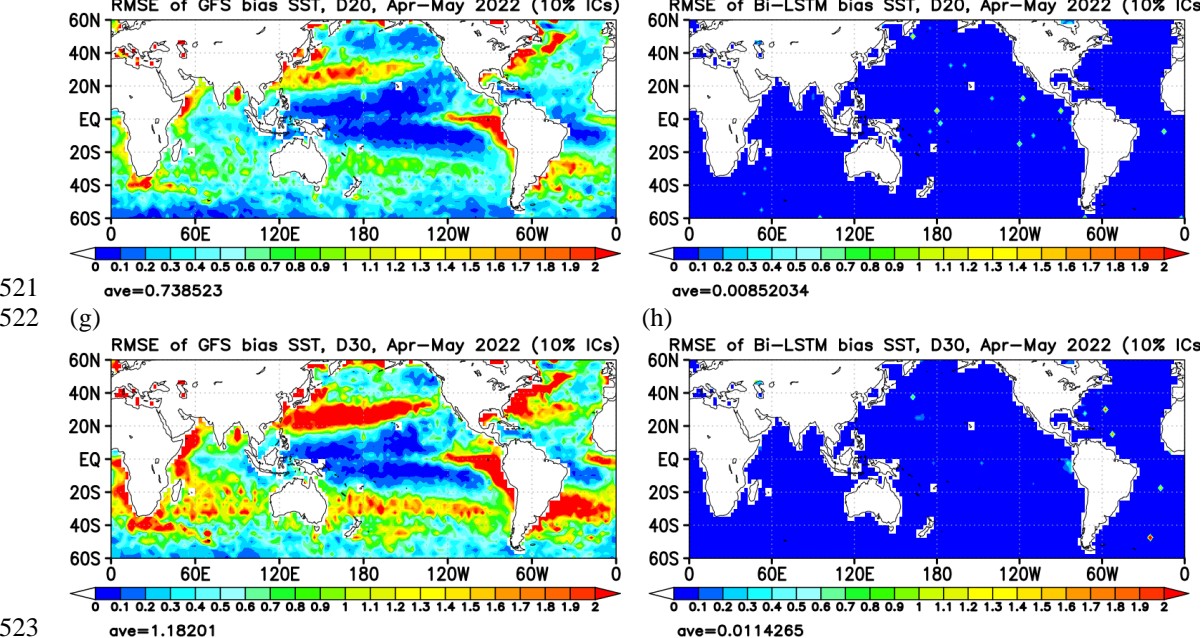


(g)                                                              (h)

**Figure A2.** The forecasting performance (RMSE) for BiasSST of the model results. The left column represents the RMSE
results from GFS, while the right column displays the RMSE results from Bi-LSTM. The figures are arranged in chronological
order from top to bottom, covering Day 5 (a, b), Day 10 (c, d), Day 20 (e, f), and Day 30 (g, h).
