# Peer review of "Enhancing Extended Weather Forecasts in the TCWAGFS Model Using Deep Learning Method for SST Bias 2 Correction"

_EGUsphere, 2025_

## Referee Comment (RC1)

**Enhancing Extended Weather Forecasts in the TCWAGFS Model Using Deep Learning Method for SST Bias Correction**

**General comments:**

The study mainly examined the TCWAGFS performance in the extended weather forecasts using several deep learning methods, especially for the improvement of SST bias correction. The topic is interesting. The research is essential as well. However, the study appears to be in an early stage of development and is not yet suitable for publication. The manuscript requires substantial revision and should be resubmitted for further consideration. The following comments highlight major concerns, among others, and suggest that the authors should restructure the manuscript and provide additional evidence to more effectively support their research objectives.

**Specific comments:**

1. For the research title, it should be: ... Deep Learning model"s" for SST Bias …
2. The abstract is not only for the results. Please reconstruct your abstract and improve the structure to what it should be.
3. Introduction to what has been done by the machine learning algorithms chosen in this research in the introduction section.
4. Introduction and note the purposes or importance of choosing TCWAGFS and SIT for this research. Briefly describe the SIT model, including its pros and cons.
5. The purpose of this research is to improve the SST forecast on the extended weather scale. However, the data are only adapted from 2021 to 2022. It is about 1.5 years. It is quite short. The experiment period should be extended to a longer period. Maybe it should be at least extended to an El Niño period (5-10 years).
6. Please make a table for the machine learning models in section 2.2. The content should include: the full names, abbreviations, references, and brief descriptions.
7. Figure 1 presents only the January forecast at day 10. Typically, the first figure should illustrate the model's baseline performance. It is recommended to include both day 1 and day 10 forecasts over the entire experimental period to provide a more comprehensive overview.
8. It is unclear whether the selected areas represent broader regions or specific points within the oceans. The presentation in Figure 1 lacks clarity and should be revised to improve its interpretability.
9. The manuscript lacks a clear explanation for the selection of the five areas

used in the demonstration. It is unclear why regions such as the Indian Ocean and the eastern coast of Australia were not included.

10. It should need to provide more physical evidence (not only statistical results) to prove that these improvements in SST forecasts really do a great job of improving the TCWAGFS forecasts at the extended weather scale, especially for each selected area.